Extended Abstract Track

# Neural Implicit Style-net: synthesizing shapes in a preferred style exploiting self supervision

**Marco Fumero**                                          FUMERO@DI.UNIROMA1.IT
**Hooman Shayani**                                HOOMAN.SHAYANI@AUTODESK.COM
**Aditya Sanghi**                                   ADITYA.SANGHI@AUTODESK.COM
**Emanuele Rodolà**                                      RODOLA@DI.UNIROMA1.IT

**Editors:** Sophia Sanborn, Christian Shewmake, Simone Azeglio, Arianna Di Bernardo, Nina Miolane

## Abstract

We introduce a novel approach to disentangle style from content in the 3D domain and perform unsupervised neural style transfer. Our approach is able to extract style information from 3D input in a self supervised fashion, conditioning the definition of style on inductive biases enforced explicitly, in the form of specific augmentations applied to the input. This allows, at test time, to select specifically the features to be transferred between two arbitrary 3D shapes, being still able to capture complex changes (e.g. combinations of arbitrary geometrical and topological transformations) with the data prior. Coupled with the choice of representing 3D shapes as neural implicit fields, we are able to perform style transfer in a controllable way, handling a variety of transformations. We validate our approach qualitatively and quantitatively on a dataset with font style labels.

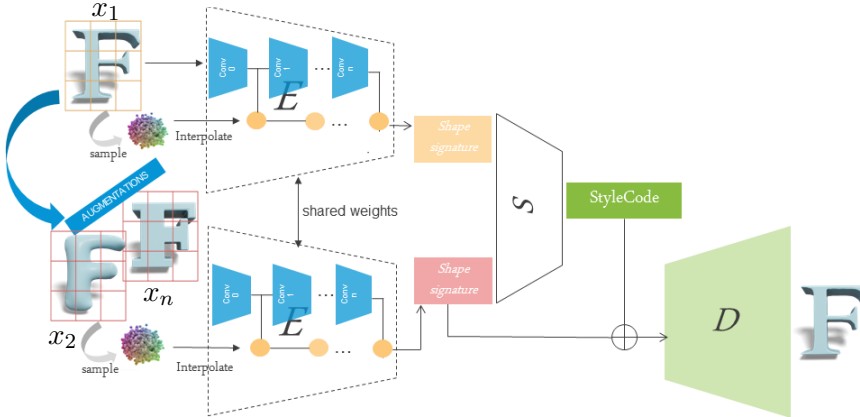

Figure 1: Model overview: an input shape $x_1$ is augmented by a set of transformations which destroy specific style features, but preserve content information, producing the augmentations $x_2, ..., x_n$. Style is encoded in a latent vector space as a nonlinear function $\mathcal{S}$ of the latent features corresponding to the input shape $x_1$ and its augmentations, respectively. The latent features are computed by a twin encoder network $E$ in a multiscale fashion. At test time the style codes condition the generation of shapes in a preferred style, using a neural implicit decoder $\mathcal{D}$.

## 1. Introduction

Being able to automatically synthesize shapes with a predefined style is a core task in computer graphics. Classical 3D style transfer techniques need to rely on a given correspondence

# Extended Abstract Track

Ma et al. (2014b), work on a fixed class of geometric or topological transformations Sorkine (2005) or use data-driven model to overcome these limitations Berkiten et al. (2017), but without explicit control on which features should be transferred, and requiring style labels or costly optimization procedures at inference time Cao et al. (2020) (see Appendix B for a complete overview). We overcome these limitations by introducing a novel approach to encode style in a compressed latent space using self supervision. Specifically, we exploit transformations of the input that, coupled with the right choice of losses, are able to disentangle style from content. This allow us to transfer style across shapes with a simple forward pass of the network at test time, by conditioning the shape generation on a given style code, where the notion of style can be defined according to the features one is interested in transferring. We choose to represent shapes as neural implicit functions to be able to learn local features that can be computed at arbitrary points in space, and be able to output shapes at arbitrary resolutions. Our contributions can be summarized as follows: (i) We observe that arbitrary sets of input augmentations are able to capture the style of a shape, disentangling it from content information in a self supervised fashion. (ii) The latent space of style codes enables multiple tasks including unsupervised style classification, style transfer, shape generation conditioned on style, and style generation.

## 2. Method

We model 3D shapes $\mathcal{M}$ as neural implicit fields $f : \mathbb{R}^3 \mapsto \mathbb{R}$, where $f(x) = \mathbb{1}_{\mathcal{M}} \min_{y \in \partial \mathcal{M}} \|x - y\|_2$, where $\mathbb{1}_{\mathcal{M}}$ is the indicator function $\mathbb{1}$ (with values 1 outside, $-1$ inside $\mathcal{M}$) and $\partial \mathcal{M}$ is the shape boundary; $f$ is approximated by a neural network. The network has the structure of an autoencoder. The encoder $E$ extracts high dimensional multi-scale local features from a regular grid of signed distance function (SDF) at specific $xyz$ locations. In detail (see Figure 1 and Appendix A.1), it works by applying 3D convolutions to the input and computing latent features at multi-scale grid points. Then taking specific point set $P_{xyz}$ as additional input, calculates the multi-scale features for the point set $P_{xyz}$ by trilinear interpolation of the latent features at the grid points. The decoder $\mathcal{D}$ takes as input the multi-scale features, as functions of the $xyz$ locations and outputs their SDF (or occupancy) values. In the feature space there is an intermediate module responsible for constructing a latent space of global style codes from multi-scale latent features of the original and augmented input shapes (see Appendix A.1). We will refer to this module as *StyleNet*. These vectors condition the decoder input, enabling to synthesize shapes according to the preferred style. The network is trained in an *unsupervised* fashion, on a reconstruction task. To characterize the functioning of the StyleNet module first we have to formalize the notion of style. As previously mentioned, there exist no universal definition for the notion of style. We choose to define style according to the following assumptions:

1. **Self consistency**: The notion of style is defined globally on the shape: the shape consistently has the same style at different spatial locations. That is the style of one region of a shape is consistent with the style of another region of the same shape.

2. **Disentanglement**: We assume that style can be *disentangled* from content, i.e. there exist transformations (e.g. smoothing, coarsening.) applied to the shapes which change and destroy the style but preserve the content of the shapes. In other words,

Extended Abstract Track

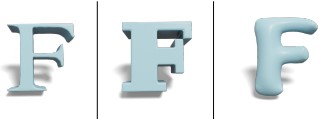

Figure 2: Examples of augmentations: the two input shapes (left column) are augmented with coarsening and smoothing transformations (center and right columns).

content is *invariant* to a predefined set of transformations. In this way we have direct control of what the notion of style is capturing.

Formally let $x_1$ be a sample shape, belonging to a specific style set $S_{x_1}$ and a specific content set $C_{x_1}$. We assume that style can be captured as a nonlinear function of the difference between high dimensional multi-scale local features, expressed as a function $E$ of the $xyz$ coordinates of $x_1$ and its transformed version $x_2 = T(x_1)$ for some transformation $T$, s.t. $T(x_1) \notin S_{x_1}$ but $T(x_1) \in C_{x_1}$. We remark that this setting is naturally extensible to handle sets of $N$ arbitrary number of augmentations $\mathcal{T} = \bigcup_{i=2}^{N+1} T_i$ for the sampled shape $x_1$, i.e. $x_i = T_i(x_1) \quad for \quad i = 2...N + 1$. Examples of these transformations are smoothing or coarsening (i.e. voxelizing and downsampling the shape at a low resolution) as shown in Figure 2. We encode the two assumptions in our model using two key inductive biases: (i) We enforce self-consistency by constructing a vector space of spatially global style codes. (ii) We enforce disentanglement in a self-supervised fashion by leveraging arbitrary input augmentations that are invariant to changes in content of the shape, incorporating consistency constraints in the network loss.

**Training Phase:** The network is trained with the following composite loss:

$$\mathcal{L}_{total} = \mathcal{L}_{SDF} + \mathcal{L}_{coarse} + \mathcal{L}_{fine} \tag{1}$$

where:

$$\mathcal{L}_{SDF} = \mathbb{E}_{(x_1,x_2)\in X}\|\mathcal{D}(E(x_2) \oplus \mathcal{S}(E(x_1), E(x_2)) - SDF_{x_1}\|_1 \tag{2}$$

$\oplus$ denoting the concatenation operator, $x_i = (x_i^{grid}, x_i^{P_{xyz}})$, for $i = 1, 2$, $SDF_{x_1}$ denoting the set of SDF values sampled near $x_1$ (i.e. $SDF(x_1^{P_{xyz}})$) and $\mathbb{E}_{(x_1,x_2)\in X}$ is the expectation over the pairs in the training distribution; and:

$$\mathcal{L}_{coarse} = \mathbb{E}_{x_2\in X}\|\mathcal{D}(E(x_2)) - SDF_{x_2}\|_1 \quad \mathcal{L}_{fine} = \mathbb{E}_{x_1\in X}\|\mathcal{D}(E(x_1)) - SDF_{x_1}\|_1 \tag{3}$$

$\mathbb{E}_{x_1\in X}$ and $\mathbb{E}_{x_2\in X}$ represent the expectation over the training distribution of $x_1$ and its augmented version $x_2$, respectively. See Figure 5-top, for a visual explanation. $\mathcal{L}_{SDF}$ is responsible for imposing disentanglement of style and content, asking to reconstruct the fine shape given the coarse features and the associated style code. However, this is not sufficient: to avoid that the conditioning information of the style code is ignored by the decoder, the consistency losses $\mathcal{L}_{coarse}, \mathcal{L}_{fine}$ penalize the reconstruction of coarse and fine shape, with zero information coming from the style code, which is replaced with a zero vector.

**Testing Phase:** At test time the style codes can be used to perform different tasks: for style transfer, a content shape $x_c$ is augmented with a transformation $\tilde{x}_c = T(x_c)$ s.t. $\tilde{x}_c \notin \mathcal{S}_c$ and its shape signature $f_c$ is computed. Then a style input shape $x_s$ is selected and its style code $v_s$ is computed. The shape resulting from the style transfer is obtained by a forward pass of the decoder $x_{c\mapsto\mathcal{S}_s} = \mathcal{D}(f_c \oplus v_s)$ (see Figure 5-bottom for a visual example).

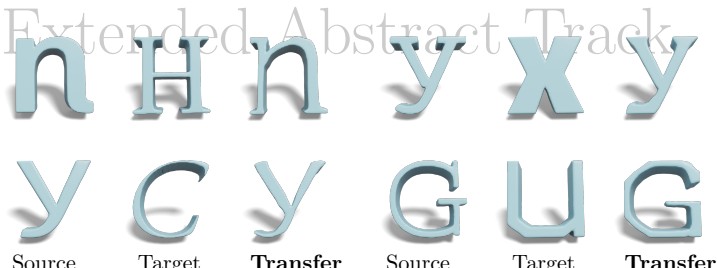

| Source | Target | **Transfer** | Source | Target | **Transfer** |

Figure 3: Examples of style transfer results.

## 3. Experiments

We evaluate our approach using the SolidLetters dataset of Meltzer et al. (2021); The dataset is made of $\approx 13k$ 3D shapes of letters represented in a large diversity of fonts, with each shape labelled with the corresponding character and font labels. As 3D datasets with labels on style are scarce and expensive to gather, we choose to use this dataset to exploit the font labels as style label, to be able to validate our approach in a quantitative way. We remark that the labels are exclusively used for evaluation, while the network is still trained in an unsupervised fashion minimizing the reconstruction loss in Eq.1. The quantitative analysis is described in section 3. See Appendix C for further results.

**Style transfer:** In Figure 3,8 we show qualitative results of style transfer experiment following the procedure described in Section 2. We selected random pairs in the dataset, (seen individually during training by the network), and performed style transfer, using coarsening and/or smoothing augmentations. In Figure 6 we show an example of aggregating style from multiple sources, where we selected many instances labelled by the same font label.

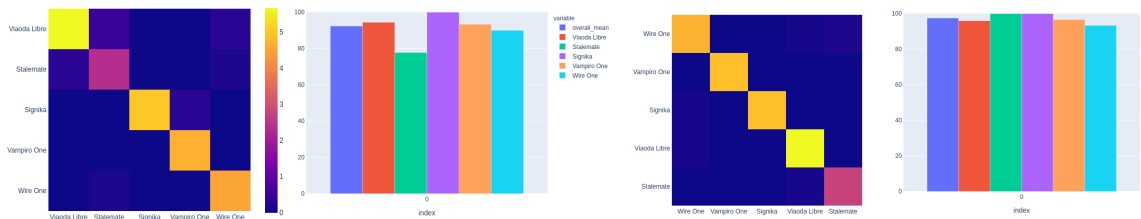

Figure 4: Linear probe experiments. *Left pair*: coarsening augmentation *Right pair*: smoothing and coarsening augmentations. For each pair: Left: confusion matrix for the style codes (closer to diagonal is better) for the selected styles; Right: accuracy scores for each font, with overall mean score reported (blue column)

**Style classification:** We selected five visually different fonts (Wire one, Vampiro, Signika, Viado Libre, Stalemate) and all the associated uppercase and lowercase letters. We trained a linear classifier on their style code space, using the ground truth font labels, with 10-fold cross validation. In Figure 4, left pair, we report the mean accuracies per font averaged across all 10 folds (right) and the confusion matrix averaged across all folds (left). We report the same statistics for a model trained with both the coarsening and smoothing augmentation on the right, showing further improvement on the accuracies. See further results on the evidence of disentanglement of style and content in Appendix C.

**Conclusions:** In this manuscript, we proposed a novel approach to perform self supervised neural style transfer on implicit representations of 3D shapes, exploiting input augmenta-

## Extended Abstract Track

tions that are able to disentangle style from content and to create a latent space of style codes. Once trained, the network can be used for different tasks such as style transfer, style classification, and style synthesis from multiple examples. We plan to further extend this work according to Appendix D.

### Acknowledgments

Marco Fumero and Emanuele Rodolà are supported by the ERC Starting Grant No. 802554 (SPECGEO) and the MIUR under grant "Dipartimenti di eccellenza 2018-2022" of the Department of Computer Science of Sapienza University

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

# Extended Abstract Track

Yongcheng Jing, Yezhou Yang, Zunlei Feng, Jingwen Ye, Yizhou Yu, and Mingli Song. Neural Style Transfer: A Review. *arXiv*, May 2017.

Sebastian Koch, Albert Matveev, Zhongshi Jiang, Francis Williams, Alexey Artemov, Evgeny Burnaev, Marc Alexa, Denis Zorin, and Daniele Panozzo. ABC: A Big CAD Model Dataset For Geometric Deep Learning. *arXiv*, Dec 2018.

Nicholas Kolkin, Jason Salavon, and Gregory Shakhnarovich. Style transfer by relaxed optimal transport and self-similarity. In *Proceedings of the IEEE/CVF Conference on Computer Vision and Pattern Recognition*, 2019.

Yanghao Li, Naiyan Wang, Jiaying Liu, and Xiaodi Hou. Demystifying neural style transfer. *arXiv preprint arXiv:1701.01036*, 2017.

Hsueh-Ti Derek Liu and Alec Jacobson. Cubic stylization. *ACM Transactions on Graphics*, 2019.

Hsueh-Ti Derek Liu and Alec Jacobson. Normal-Driven Spherical Shape Analogies. *arXiv*, Apr 2021.

Hsueh-Ti Derek Liu, Vladimir G. Kim, Siddhartha Chaudhuri, Noam Aigerman, and Alec Jacobson. Neural subdivision, 2020.

Jikai Liu and Yongsheng Ma. A survey of manufacturing oriented topology optimization methods. *Advances in Engineering Software*, 100, 2016. ISSN 0965-9978.

Zhaoliang Lun, Evangelos Kalogerakis, and Alla Sheffer. Elements of style: Learning perceptual shape style similarity. *ACM Trans. Graph.*, 34(4), jul 2015. ISSN 0730-0301.

Zhaoliang Lun, Evangelos Kalogerakis, Rui Wang, and Alla Sheffer. Functionality preserving shape style transfer. *ACM Trans. Graph.*, 35(6), nov 2016. ISSN 0730-0301.

Chongyang Ma, Haibin Huang, Alla Sheffer, Evangelos Kalogerakis, and Rui Wang. Analogy-driven 3D style transfer. *Computer Graphics Forum*, 33(2), 2014a.

Chongyang Ma, Haibin Huang, Alla Sheffer, Evangelos Kalogerakis, and Rui Wang. Analogy-driven 3d style transfer. In *Computer Graphics Forum*, volume 33, 2014b.

Riccardo Marin, Arianna Rampini, Umberto Castellani, Emanuele Rodolà, Maks Ovsjanikov, and Simone Melzi. Instant recovery of shape from spectrum via latent space connections, 2020.

Peter Meltzer, Hooman Shayani, Amir Khasahmadi, Pradeep Kumar Jayaraman, Aditya Sanghi, and Joseph Lambourne. Uvstyle-net: Unsupervised few-shot learning of 3d style similarity measure for b-reps. *arXiv preprint arXiv:2105.02961*, 2021.

Lars Mescheder, Michael Oechsle, Michael Niemeyer, Sebastian Nowozin, and Andreas Geiger. Occupancy networks: Learning 3d reconstruction in function space, 2019.

Youssef Mroueh. Wasserstein Style Transfer. In *International Conference on Artificial Intelligence and Statistics*. PMLR, Jun 2020.

# Extended Abstract Track

Jeong Joon Park, Peter Florence, Julian Straub, Richard Newcombe, and Steven Lovegrove. Deepsdf: Learning continuous signed distance functions for shape representation, 2019.

Charles R. Qi, Hao Su, Kaichun Mo, and Leonidas J. Guibas. PointNet: Deep Learning on Point Sets for 3D Classification and Segmentation. *arXiv*, Dec 2016.

Mattia Segu, Margarita Grinvald, Roland Siegwart, and Federico Tombari. 3dsnet: Unsupervised shape-to-shape 3d style transfer. *arXiv preprint arXiv:2011.13388*, 2020.

Olga Sorkine. Laplacian Mesh Processing. In Yiorgos Chrysanthou and Marcus Magnor, editors, *Eurographics 2005 - State of the Art Reports*, 2005.

Yu Wang and Justin M. Solomon. Intrinsic and extrinsic operators for shape analysis. *Handbook of Numerical Analysis*, 2019.

## Appendix A. Model architecture

### A.1. Model architecture

Input shapes are preprocessed computing a grid of SDF values at a given resolution (typically $32^3$ to $128^3$ ), and sampling a point cloud $xyz$ near the surface, computing its SDF values. The grid and the $xyz$ coordinates are given to the network as input, while the ground SDF values at $xyz$ are used for training the network on reconstruction. To refer to the sdf grid and the sampled point cloud specific for a shape $x_1$ we use the following notation: $x_1^{grid}, x_1^{P_{xyz}}$. The model architecture is depicted in Figure 1. The network takes as input pair of shapes $(x_1, x_2)$ where $x_2 = T(x_1)$ for some transformation $T$. Depending on the definition of style, the transformation $T$ changes the style but is invariant to the content of the shape.

The encoder $E$ computes an overcomplete feature description of the point cloud sampled near the surface, for the shape $x_1$ and its transformed version $x_2$. This description which we refer to as *shape signature* is obtained in the following way: the uniform grid of SDF values is fed to 3D convolutional blocks which are followed by max pooling layers; (similar to Chibane et al. (2020)), from each feature grid at a different resolution (corresponding to the output of convolution+maxpooling block), we sample points at specific locations corresponding to the points sampled near the surface, using trilinear interpolation. These are passed through the decoder $\mathcal{D}$ (a simple MLP with LeakyRelu activation functions) which outputs the SDF value at the corresponding $xyz$ location. The intermediate StyleNet module, (detailed in the inset on the right) denoted

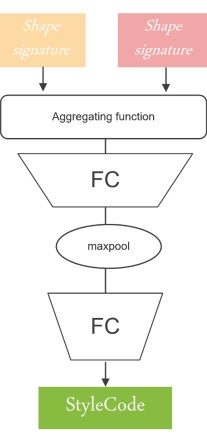

with $\mathcal{S}$, takes the shape signature of $x_1$ and $x_2$ as input, and is responsible to output a global latent vector, representative of the style of $x_1$, given $x_2$. It works by combining the two shape signatures with some aggregation function ( in the experiments we used simple vector subtraction, but other solution, e.g. an MLP, are possible). The output of the aggregation function is then passed it through: (i) a MLP network which is responsible for mixing the features; (ii) a maxpool layer, which reduces over the number of points dimension; (iii) a final MLP layer to reduce the dimensionality and obtain the output global vector which is descriptive of the style of $x_1$.

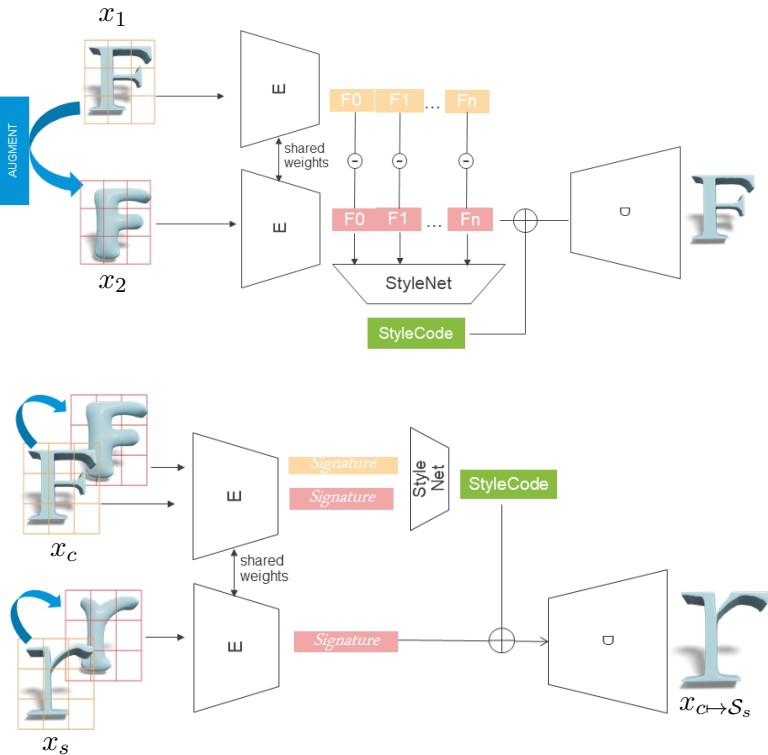

Figure 5: *Top*: during training phase the network is enforced to reconstruct the original input from the augmented one, giving structure to the latent space of style codes; *Bottom:* at test time, the latent space of style vectors can be exploited for downstream tasks, such a style transfer: here the augmented "r" input is upsampled according to the style code computed from "F" and its augmentation.

**Experimental settings:** For each shape in the dataset, we precompute grids of SDF values with resolution $64^3$, and store approximately 100k points near the surface together with their SDF values, to enable fast dataloading during training. The points $\hat{P}_{xyz}$ are sampled according to:

$$\hat{P}_{xyz} = P_{xyz} + \mathcal{N}(0, \sigma I)$$

where $P_{xyz}$ is a randomly sampled point on the surface and $\sigma$ is chosen to be equal to 0.01 (multiple values for sigma can be used for datasets with more detailed shapes). During training $N = 2048$ points are uniformly and randomly sampled from the stored ones, for each shape in a minibatch. The dataset is augmented with the following set of transformations: coarsening (i.e. voxelizing to a resolution of $8^3$ and then recomputing the $64^3$ grid on the resulting shape) and smoothing using a variant of laplacian smoothing with less shrinkage Barroqueiro et al. (2021). We experimented with two different settings, one with only the coarsening augmentations and the other with smoothing and coarsening. The architecture composition for this experiments is detailed in Table 1.

Extended Abstract Track

| $E$ | $\mathcal{D}$ | $\mathcal{S}$ |
|---|---|---|
| Conv3D(16,3,1);LReLU() | FC(code+fsize,256) | FC(fsize,fsize) |
| MaxPool(2) | LReLU() | LReLU() |
| Conv3D(32,3,1);LReLU() | FC(256,256) | FC(fsize,fsize) |
| Conv3D(32,3,1);LReLU() | LReLU() | LReLU() |
| MaxPool(2) | FC(256,256) | FC(fsize,fsize) |
| Conv3D(64,3,1);LReLU() | LReLU() | Maxpool(pt) |
| Conv3D(64,3,1);LReLU() | FC(256,1) | FC(fsize,256) |
| MaxPool(2) | | LReLU() |
| Conv3D(128,3,1);LReLU() | | FC(256,256) |
| Conv3D(128,3,1);LReLU() | | LReLU() |
| MaxPool(2) | | FC(256,csize) |
| Conv3D(128,3,1);LReLU() | | |
| Conv3D(128,3,1) | | |

Table 1: Architecture: the specific modules used in the experiments are reported for the encoder $E$, the decoder $\mathcal{D}$, the StyleNet module $\mathcal{S}$ with the overall feature size $fsize := 1 + 16 + 32 + 64 + 128 + 128 = 369$, the size of the style code $csize := 256$ and $pt$ being the number of sampled points fed to the network.

## Appendix B. Related work

### B.1. 2D neural style transfer

In the image domain neural style transfer was introduced in Gatys et al. (2016) as an optimization problem where, given a *content* image and a *style* image, the objective is to minimize the L2 distance between the Gram matrices of their features at specifically selected layers of a CNN vision architecture. Later in Li et al. (2017) this was shown to be equal to a distribution alignment problem between features, where the style loss in Gatys et al. (2016) was shown to be equal to align the second moments of the feature distributions of the two input images. This lead to an entire line of works Kolkin et al. (2019); Huang and Belongie (2017); Jing et al. (2017) exploiting different techniques, such as optimal transport Mroueh (2020) to solve the distribution alignment problem.

### B.2. Neural implicit fields

3D shapes can be represented by a neural network implicitly that aims to predict either binary occupancy values or signed distance function (SDF) values at continuous $xyz$ locations Park et al. (2019); Chen and Zhang (2019); Gropp et al. (2020); Mescheder et al. (2019). We chose to work with this representations as it provides a discretization-free representation of the shape, it can well handle topological variations, and is suitable for integration with topology optimization Liu and Ma (2016) pipelines.

### B.3. 3D style transfer

In the context of 3D shapes, style transfer has been classically tackled as a two stage problem: first capture informative signals about the local geometry and the structural similarity between a set of shapes (i.e. get a *measure* of style) and then transport extracted features (*transferring* the style), exploiting some pairwise point correspondence Lun et al. (2015, 2016); Ma et al. (2014a). The principal limitations of these methods is that the correspon-

# Extended Abstract Track

dence must be known, and the shape optimization afterwards is usually slow. Spectral techniques Sorkine (2005); Wang and Solomon (2019), exploit the eigenbasis of geometry informative operators (e.g. the Laplace-Beltrami operator) to solve for a functional correspondence between shapes and then perform frequency transfer . The principal limitation of these methods is that there the notion of style is reduced to the high frequency features in the spectrum of the operator. These techniques were extended with learning based approaches in Berkiten et al. (2017); Marin et al. (2020). In Liu and Jacobson (2019, 2021) style transfer is recasted as an optimization problem in the normal domain, to deform a shape to align to a specific geometric primitive such as the cube. While it works nicely, the stylization is limited to low frequency deformation, which can align the shape exclusively to the target primitive. More recently, style transfer has been tackled with learning based methods. This benefited both, the ability to capture and measure the style signal Meltzer et al. (2021) by exploiting the deep features extracted from the data, and the transferring the style by solving implicitly for the correspondence between shapes, e.g. by recasting the problem to a generative model conditioned on style information. Cao et al. (2020) extended the approach of Gatys et al. (2016) to Poincloud data, using a PointNet encoder Qi et al. (2016). Segu et al. (2020) extended this work assuming disentanglement between content and style. In Liu et al. (2020) it was observed that by overfitting a network to learn a subdivision scheme for meshes, that network could be used at test time to stylize another low resolution mesh. Our method differs from these in that it is tailored to work with neural implicit representations, enables style transfer with a simple forward pass, without the requirement of solving any optimization procedure, and can capture multiple complex styles (and synthesize new ones) without the need to retrain the network from scratch. Closer to our work in Chen et al. (2021b) the authors proposed a deep generative model that refines a low-resolution coarse voxel shape, via voxel upsampling, into a detailed higher resolution shape, by conditioning on a style code. This work differs in the fact that we work with neural implicit representations, therefore alleviating the problem of working with higher resolutions of voxels, and their work is specific to voxel upsampling, while ours can handle different input transformations.

## Appendix C. Additional results

**Aggregating style from a set of shapes:** Our method makes it also possible to synthesize a single style code from a collection of shapes that essentially come from the same style. Different letters from the same font is a clear example of such situation when different shapes share the same style but no single shape can contain all the stylistic features required to define that style. Given a set of shapes $X = x_1..x_n$, respectively belonging to the set of styles $\mathcal{S}_1..\mathcal{S}_n$ we can compute their shape signatures $f_1...f_n$ and concatenate them, before feeding them to the StyleNet module. In this way one can get a single style code which is representative of the overall style of a collection of shapes (see Figure 6).

**Style clusters:** Our network is able to correctly disentangle style information from content. This can be seen cluster different styles in the latent space of codes in an unsupervised fashion, without relying on any style label. In Figure 7 we visualize TSNE projections cor-

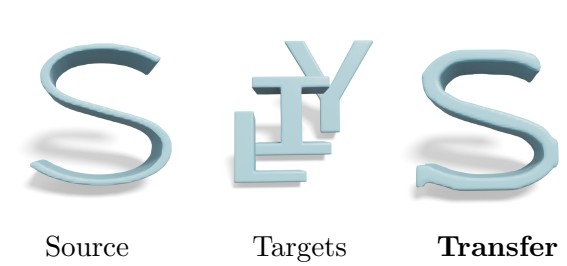

Source    Targets    **Transfer**

Figure 6: Style transfer from multiple sources: For a content source shape (left) we can select multiple style targets (center) and obtain a single style code to generate the transfer result on the right.

responding to the style codes in the quantitative experiment in Figure 4 showing further evidence of the disentanglement properties of the style space.

**Additional style transfer results:** For additional qualitative examples of style transfer refer to Figure 8.

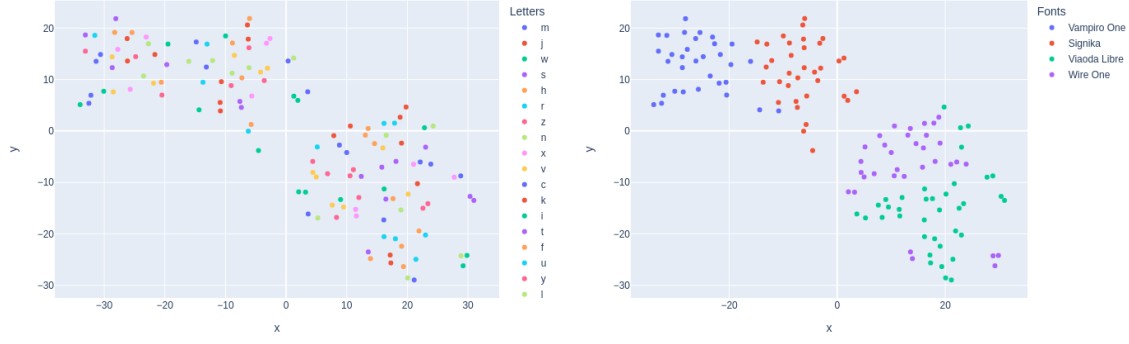

Figure 7: TSNE projections of style space: *Left:* TSNE projections of style codes labeled by letter, *Right:* TSNE projections of style codes labeled by style. While the space does not show any structure w.r.t to letters labels (i.e. content), it shows clustering w.r.t. to style labels, assessing the disentanglement capabilities of our approach.

## Appendix D. Conclusions and Future work

**Future work:** We performed an experimental preliminary study on the font labelled dataset Meltzer et al. (2021). Nevertheless we plan to extend these results to real datasets such as Koch et al. (2018); Chang et al. (2015) and to test the integration of the method with topology optimization pipelines. From a methodological perspective, multiple paths could be investigated: performing multiresolution style transfer building a hierarchy of style codes, extending the architecture akin to Chen et al. (2021a), adding a Gaussian prior to

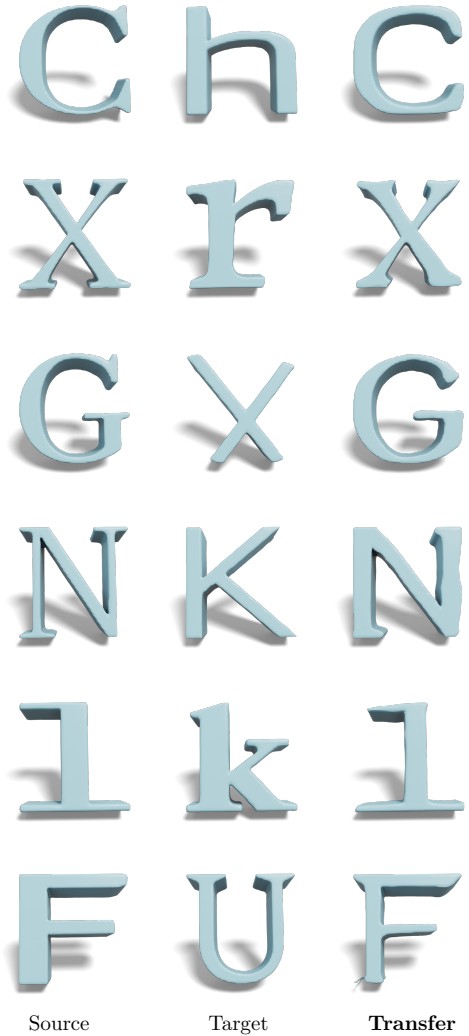

Source      Target      **Transfer**

Figure 8: Examples of style transfer results. From left to right, content input, style target, and transfer result

the latent space of codes to be able to generate novel styles from scratch by sampling at test time.

