# OpenReview forum: "Neural Implicit Style-net: synthesizing shapes in a preferred style exploiting self supervision"
_NeurIPS.cc/2022/Workshop/NeurReps — NeurReps 2022 Poster_

### Official Review · Reviewer_qw9c · 2022-10-10
**A new style transfer method on SDF**

**Confidence:** 5
**Soundness:** 4
**Presentation:** 3
**Contribution:** 3
**Overall Rating:** 8

**Summary:**

The paper proposes a new approach to disentangle style from 3D shapes represented as implicit neural fields. The style is defined by inductive biases that allow training the method in a self-supervised manner.
The model is composed of three parts. An encoder computes a multi-scale shape signature from a regular grid of signed distance function at a specific point set. Then, an internal StyleNet extracts a StyleCode from the input shape's signature and the augmented shape combination. Finally, the decoder concatenates the shape signature and the StyleCode to generate a new 3D shape.
The method is tested on style transfer and classification. The results demonstrate the effectiveness of the new model.


**Questions:**

It is unclear how you handle the shape signature when you have more than one augmentation. Do you aggregate all the augmented signatures $x_s^n$ to the signature of $x_c$? For instance, do you subtract all $x_s^n$ to $x_c$?

Did you try any other aggregation other than subtraction? Is it possible to define an algebra between shape signatures to obtain new combinations of style codes?

At the end of page 3, some parts are unclear. The last line before “Testing Phase” has a point in the middle of the sentence.
In Equation 3, I would add the concatenation to the zero vector in the input of $\mathcal{D}$.
In the Testing Phase section, there may be some typos:
- $ x_c \in \mathcal{S}_c$ should have $ \tilde{x_c} $
- $ \mathcal{D}(f_c \bigodot v_s)$, $\bigodot$ is not define. It may be a typo with $\bigoplus$.
- The reference to Figure 5 should be to the bottom and not right.

**Limitations:**

The authors do not discuss the limitations of their work.
For instance, would it be possible to modify only a specific region of a shape?


**Recommended Decision:**

3: Accept

**Relevance:**

3: Solid fit

**Strengths And Weaknesses:**

I consider the work novel. I appreciate the idea of performing style manipulation on neural implicit fields.

The experiments (both qualitative and quantitative) demonstrate the efficacy of the method. Moreover, the resulting shapes have a fine style fidelity, especially between pairs never seen at training time.

The paper is well written. There are some typos and clarifications that I reported in the Questions.

I think the work is of interest to this community. The paper shows promising results and interesting future directions.

**Submission Track:**

Extended Abstract (4 Page)

---

### Official Review · Reviewer_HqsQ · 2022-10-13
**Disentangling Style from the content and using it for style transfer**

**Confidence:** 4
**Soundness:** 4
**Presentation:** 4
**Contribution:** 4
**Overall Rating:** 6

**Summary:**

This paper presents a way to disentangle style from the content of the image in a self supervised manner, and using that for style transfer in an unsupervised manner. The method the paper provides includes some strong on assumptions on how a 'style' should be defined as. The authors defines style as followed :
1. **Self Consistency** : The style of the shape is same whether you look at it as a whole or in some small region.
2. **Disentanglement** : Content of the shape is preserved even if style is changed/destroyed. (Hence you can learn it).

The proposed model has two very different phases (training and testing phases). It leverages a twin encoder to compute the latent features (in this case style). The phases are described below :
- **Training Phase** : Force the network to reconstruct the original input under different kinds of augmentations. Hopefully this allows the latent representation to be forced to the style of the original image.
- **Testing Phase** : Using the latent representation for style transfer.

The method proposed in this paper aims to provide a way to disentangle the style from shapes, as well allow the learned latent representation to be used for other tasks like style transfer. The proposed method is also faster than previous methods for style transfer for inference, as only one forward pass is required.

**Questions:**

1. We can see that the proposed method is faster, but what about quality of reconstructions? Does this method outperform in that domain too? I understand this is an early work but I was just curious.
2. Does this method generalize to other types of style transfer categories like paintings etc? Do you have any results regarding those?

**Limitations:**

The authors clearly discussed their limitations and future directions in the appendix.

**Recommended Decision:**

3: Accept

**Relevance:**

3: Solid fit

**Strengths And Weaknesses:**

**Strengths** : I think this paper is very will written and easy to understand in most parts. The proposed method seems like a novel method to force the latent representation to the style of an image and demonstrates clear relation between the problem they are trying to solve and the solution. The claims seems to be very well supported with different kinds of experiment studies in the paper.

**Weaknesses** : I do think some parts of the paper are confusing (however, the appendix is helpful so Kudos to that). I really recommend moving Figure 5 to the main paper as that was the crux of the paper. The paper also said that the other methods were slow at inference and their own method only requires one forward pass, but I feel like it is a bit lacking in comparison with some state of the art style transfer methods in the experiments section. That would make their claim stronger.

**Submission Track:**

Extended Abstract (4 Page)

---

### Official Review · Reviewer_y7Tq · 2022-10-14
**Interesting paper about neural style transfer for 3D shapes**

**Confidence:** 3
**Soundness:** 3
**Presentation:** 3
**Contribution:** 2
**Overall Rating:** 6

**Summary:**

This submission proposed a neural style transfer method for 3D shapes, specially, the shape is represented as neural implicit fields.

1. In the training stage,  the network is enforced to reconstruct the original input from the augmented one, giving structure to the latent space of style codes.
2. In the testing stage,  given a target shape, the style code is computed and then we can get the transferred shape from a forward pass.

**Questions:**

How much time would it take for the training stage? It seems that optimizing the loss in this paper is not trivial.

**Limitations:**

Examples in this submission is relative easy, it would be very interesting to extend this submission to a paper by adding more complicated examples.

**Recommended Decision:**

3: Accept

**Relevance:**

3: Solid fit

**Strengths And Weaknesses:**

Strengths: Classical 3D style transfer techniques need to rely on a given correspondence, this paper overcomes this problem by introducing a novel approach to encode style in a compressed latent space using self supervision, which makes it possible to transfer style across shapes with a simple forward pass of the network at test time.

Originality issue: using neural implicit fields to represent 3D shapes is not new.

**Submission Track:**

Extended Abstract (4 Page)

---

### Decision · Program_Chairs · 2022-10-21

Accept (Poster)